# Multi-Level Contrastive Learning for Dense Prediction Task

## Abstract

In this work, we present Multi-Level Contrastive Learning for Dense Prediction Task (MCL), an efficient self-supervised method for learning region-level feature representation for dense prediction tasks. Our method is motivated by the three key factors in detection: localization, scale consistency and recognition. To explicitly encode absolute position and scale information, we propose a novel pretext task that assembles multi-scale images in a montage manner to mimic multi-object scenarios. Unlike the existing image-level self-supervised methods, our method constructs a multi-level contrastive loss that considers each sub-region of the montage image as a singleton. Our method enables the neural network to learn regional semantic representations for translation and scale consistency while reducing pre-training epochs to the same as supervised pre-training. Extensive experiments demonstrate that MCL consistently outperforms the recent state-of-the-art methods on various datasets with significant margins. In particular, MCL obtains 42.5 $AP^{bb}$ and 38.3 $AP^{mk}$ on COCO with the 1x schedule fintuning, when using Mask R-CNN with R50-FPN backbone pre-trained with 100 epochs. In comparison to MoCo, our method surpasses their performance by 4.0 $AP^{bb}$ and 3.1 $AP^{mk}$. Furthermore, we explore the alignment between pretext task and downstream tasks. We extend our pretext task to supervised pre-training, which achieves a similar performance to self-supervised learning. This result demonstrates the importance of the alignment between pretext task and downstream tasks, indicating the potential for wider applicability of our method beyond self-supervised settings.

## 1 Introduction

A generic large-scale supervised pre-training is a critical auxiliary task for the computer vision community to progress, like ImageNet(Deng et al., 2009) pre-training, which has been validated by many works (Erhan et al., 2010; He et al., 2019; 2017; Lin et al., 2017; Qiao et al., 2021; Ren et al., 2015; Sohn et al., 2020). The benefits of initializing the model with pre-trained weights include faster convergence and better generality for downstream tasks. Recently, self-supervised learning (SSL) based on instance discrimination tasks has driven many advances, achieving state-of-the-art results on the challenging ImageNet dataset under the $k$-NN and linear probing evaluation policy. Despite their advanced performance on classification tasks, some recent works (Pinheiro et al., 2020; Wang et al., 2021; Wei et al., 2021; Xie et al., 2021b; Yang et al., 2021) have pointed out a common weakness of these methods: The image-level representation learning does not transfer well to dense prediction tasks, such as object detection and instance segmentation. Additionally, the success of state-of-the-art methods (Caron

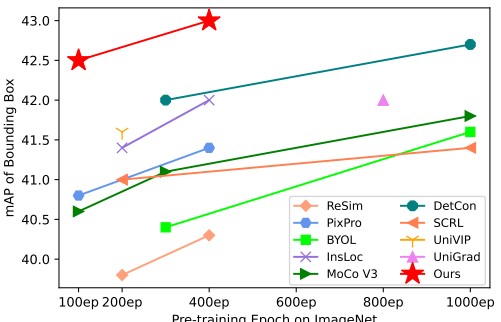

Figure 1: Our efficient self-supervised learning method, MCL, consistently outperforms the previous state-of-the-art methods on the COCO detection downstream task, while significantly reducing the pre-training epochs. MCL pre-trained on ImageNet for 100 epochs obtains 42.5 $AP^{bb}$ and 38.3 $AP^{mk}$ on COCO dataset with the standard 1x schedule (Wu et al., 2019). Transfer performance is measured by finetuning a Mask-RCNN with ResNet50-FPN backbone.

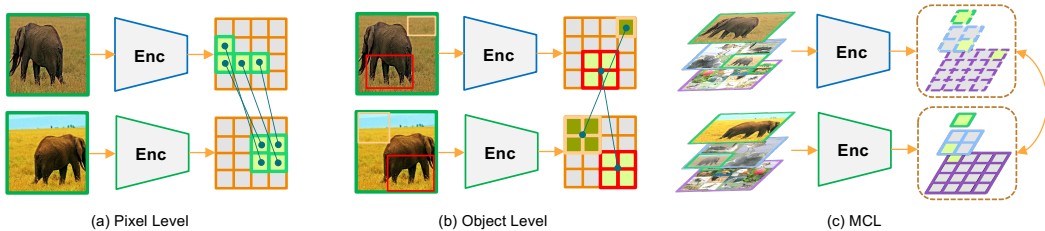

Figure 2: (a) Pixel-level methods match positive feature pairs based on the transportation cost of feature distance, which does not guarantee precise assignment. (b) Object-level methods obtain localization by off-the-shelf algorithms, whose predictions are low-quality on a non-object-centric dataset. (c) Our method assembles multiple augmented views with different sizes in a montage manner and constructs multi-level contrastive learning on each sub-image. As a result, MCL learns regional representation for precise localization, scale consistency among multi-scale crops and semantic representation that generalizes well on the dense prediction tasks.

et al., 2020; Chen et al., 2020a; Grill et al., 2020; He et al., 2020; Hénaff et al., 2021) requires several times more training epochs than supervised pre-training.

In contrast to the ImageNet classification task, where objects have a small variation in scale, object detection datasets typically have a large scale variation across object instances, and precise localization of objects is required. Therefore, an ideal detector is supposed to be scale-consistent across object instances and encode position information accurately. Pixel-level SSL methods (Liu et al., 2020; Wang et al., 2021) consider the spatial structure information as shown in Fig. 2(a). These pretext tasks treat each pixel in an image as a single instance and encourage the model to distinguish each pixel from others within the image. Unfortunately, the matching rule of positive pixel pair is based on the transportation cost of feature distance, which does not guarantee precise and stable feature target assignment. Object-level SSL methods (Hénaff et al., 2021; Wei et al., 2021) focus on the proposals from some off-the-shelf algorithms, such as Selective Search (Uijlings et al., 2013) and Multiscale Combinatorial Grouping (Arbeláez et al., 2014), as illustrated in Fig. 2(b). However, the bounding box and segmentation mask proposals are not accurate enough on the non-object-centric dataset, such as COCO. The low-quality proposals yield an inferior result for the downstream dense prediction tasks due to the localization noise.

Our proposed method, Multi-Level Contrastive Learning (MCL), is motivated by the key factors of detection: localization, scale consistency, and recognition. MCL is a novel and highly efficient self-supervised learning framework for dense prediction tasks, which has achieved state-of-the-art transfer performance on downstream tasks while significantly reducing the training epochs. To achieve localization, scale consistency, and recognition, we design a montage assembly that assembles multi-scale images into a non-overlapping grid, mimicking multi-object scenarios. The montage assembly explicitly encodes the position and scale information of images. Additionally, we adopt a scale-aware positive target assignment strategy on the feature pyramid, which produces a multi-scale feature representation with strong semantic information. Moreover, MCL proposes a series of contrastive modes to improve scale consistency. Each component image in the montage image is treated as an independent instance, and features are accurately extracted based on image coordinates. This bridging of the gap between the pretext task and downstream tasks ensures accurate feature target assignment. To further investigate the alignment between pretext task and downstream tasks, we extend our pretext task to supervised pre-training. Our supervised pre-training achieves similar performance to self-supervised pre-training, demonstrating the importance of task alignment.

To evaluate MCL, we conduct extensive experiments on benchmarks for various dense prediction tasks. As shown in Fig. 1, MCL achieves state-of-the-art transfer performance pretrained on the ImageNet dataset, while significantly reducing the training cost to 100 epochs on ImageNet. MCL pre-trained on ImageNet for 100 epochs obtains 42.5 $AP^{bb}$ and 38.3 $AP^{mk}$ on the COCO dataset with the standard 1x schedule (Wu et al., 2019) and surpasses MoCo by 4.0 $AP^{bb}$ and 3.1 $AP^{mk}$, using Mask R-CNN with R50-FPN backbone. MCL pre-trained on the unlabeled COCO dataset also achieves a state-of-the-art result, 41.8 $AP^{bb}$ and 37.7 $AP^{mk}$. This result shows that MCL benefits from the carefully designed multi-level pretext task rather than the dataset bias (Purushwalkam & Gupta, 2020).

Our contributions are listed as follows: (1) An efficient self-supervised method, Multi-level Contrastive Learning, is designed to align the pretext task with the dense prediction tasks, improv-

ing scale invariance and localization precision. (2) Montage assembly is introduced in the self-supervised learning field for **the first time** to construct a montage image, mimicking multi-scale multi-object scenarios, with no need of pseudo label boxes. (3) Our method achieves **state-of-the-art** transfer performance on the dense prediction downstream tasks, such as detection, segmentation, and pose estimation while reducing the pre-training cost to 100 ImageNet epochs.

## 2 RELATED WORK

**Instance contrastive learning.** Instance-level contrastive learning considers each image as a singleton, only one sample in a class (Bojanowski & Joulin, 2017), which considers two augmented views of the same image as positive to be pulled closer, and all other images negative to be pushed further apart. MemoryBank (Wu et al., 2018) stores previously-computed representation in a memory bank to compare instances based on noise contrastive estimation. MoCo (He et al., 2020) uses a momentum encoder to store representation in a temporal manner, allowing the dictionary to be large. SimCLR (Chen et al., 2020a;b) shows that the memory bank is not necessary when the mini-batch size is large enough. SwAV (Caron et al., 2020) clusters the data while enforcing consistency between cluster assignments. Besides, SwAV adopts multi-crop data augmentation, which uses a mix of views with different resolutions in place of two full-resolution views. BYOL (Grill et al., 2020) and SimSiam (Chen et al., 2020a) explore directly maximizing the similarity between two views of one image without negative pairs. Despite the success of instance-level contrastive learning on ImageNet linear probing, instance-wise contrastive learning does not encode position information explicitly, treating all regions equally. In contrast, MCL views each subimage in the montage image as a singleton to explicitly encode image localization with high fidelity.

**Dense Representation Learning.** Dense representation learning predicts at the pixel level, compared with the instance contrastive learning. Recently, some self-supervised learning methods that learn at pixel level representation are proposed. ULDVR(Pinheiro et al., 2020) learns pixel-wise representation by forcing local features to remain constant over different view conditions. DCL(Wang et al., 2021) optimizes a pairwise contrastive similarity loss at the pixel level between two views of input images by the Hungarian matching strategy. Self-EMD(Liu et al., 2020) shares a similar basic idea, but updates the matching strategy to Earth Mover's distance(Rubner et al., 2000). PixPro (Xie et al., 2021b) matches the feature pixels by a hand-crafted decision rule. These matching strategies only implicitly map the localization in feature maps to the euclidean coordinate in the input image, but do not guarantee a precise feature target assignment. Different from the pixel-level label assignment, MCL assigns a positive sample by matching the image regions with different sizes in the montage image on multi-level feature maps. Our scale-aware assignment strategy ensures the precision localization of each feature point.

**Object-Level Representation Learning.** Both single-stage detectors and two-stage detectors attend to a manageable number of candidate object regions and evaluate convolutional networks on each region. The regions of interest have rectangular shapes and come in different sizes. RoIAlign (He et al., 2017) is proposed to extract the features of particular regions on the convolutional feature maps. Following Fast-RCNN (Girshick, 2015), SoCo (Wei et al., 2021) selects the proposal bounding boxes generated from Selective Search (Uijlings et al., 2013) and applies RoIAlign to extract object features by constructing multiple augmented views, which is used for contrastive loss. DetCon (Hénaff et al., 2021) identifies object-based regions with the off-the-shelf approximate segmentation algorithms (Arbeláez et al., 2014; Felzenszwalb & Huttenlocher, 2004) to produce a semantic mask. The contrastive detection objective then pulls together pooled feature vectors from the same mask and pushes apart features from different masks and different images. Whereas, the segmentation mask and bounding box predicted by the off-the-shelf methods are not accurate enough, incurring pseudo-label noise, which leads to an inferior result on the non-object-centric dataset. In contrast, MCL constructs montage images and precisely annotates the localization of each component image. As a result, MCL maintains a high transfer performance when pre-trained on the COCO dataset.

## 3 METHOD

Our goal is to learn regional semantic representation and scale consistency without supervision while keeping the training epoch reasonable. While instance-level SSL methods (He et al., 2020; Misra & Maaten, 2020) have shown success in learning occlusion-invariant representations (Purushwalkam

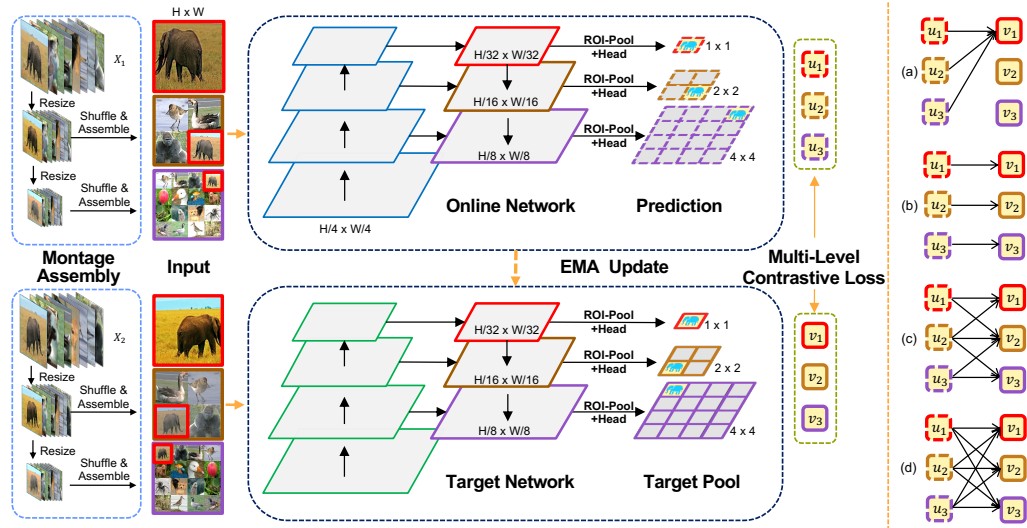

Figure 3: Overview of our method. This figure illustrates MCL with a model, whose FPN contains 3 levels. The image batch $X$ is processed via the same augmentation pipeline with different random seeds. The images are downsampled by a factor of 2 and shuffled to construct montage input. The subfigures are multiscale and precisely localized. The feature pyramid is further ROI-pooled according to the subfigure location. Contrastive learning is performed on multi-level features to learn semantic regional representations via scale consistency regularization. $u_i$ and $v_i$ are extracted from the $i$-th level feature pyramid. The arrow means to set the corresponding target feature as the positive sample and the other target features at the same level as the negative samples. The target network is not optimized by gradient and updated by the online network in an EMA manner.

& Gupta, 2020), we present a new approach that applies instance discrimination at the region level for learning visual representations that generalize well to dense prediction tasks. As illustrated in Fig. 3, our method, MCL, constructs multiple augmented views in different sizes and produces a multi-scale feature pyramid, on which a contrastive loss is applied across the levels. The montage assembly guarantees the precision of pseudo bounding box label, which explicitly encodes the absolute position information. To align the pretext task and downstream tasks, all the network modules in the downstream model are pre-trained for a well-initialized representation. We also extend our pretext task to supervised pre-training and provide more details on this in Appendix.

## 3.1 MONTAGE ASSEMBLY

The photomontage is the process and the result of making a composite photograph by cutting, gluing, rearranging, and overlapping two or more photographs into a new image. An interesting observation is that montage process assembles images in different scales at specific locations, which explicitly encodes the position and scale information. In our approach, the image batch $X$ is processed through the same augmentation pipeline with different random seeds. To handle the large scale variation, the augmented images are downsampled to $\frac{1}{2^s}$ of the original size, where the $s$ ranges from $\{0, 1, 2, ..., S-1\}$ and matches the level of feature maps in FPN. To encode position and scale information, all the downsampled images with the same downsampling ratio are randomly combined to construct a montage image. The resulting montage images have aligned shapes with the original augmented images. The pseudo code for this process is provided in Alg. 1 for clarity.

**Algorithm 1** Montage Pseudo Code

```
# s: the level of downsampling ratio
# x: the input images batch with shape of (B, C, H,
    W)
ratio = pow(2, s)
x_aug = aug(x)  # data augmentation
x_aug_ds = interpolate(x_aug, scale_factor = 1. /
    ratio)
x_aug_ds = shuffle(x_aug_ds)
B_ds = B / ratio / ratio
H_ds, W_ds = H / ratio, W / ratio
x_aug_ds = x_aug_ds.reshape(B_ds, ratio, ratio, C,
    H_ds, W_ds)
x_aug_ds = x_aug_ds.permute(0, 3, 1, 4, 2, 5)
x_aug_ds = x_aug_ds.reshape(B_ds, C, H, W)
```

## 3.2 MULTI-LEVEL CONTRASTIVE LEARNING

Detectors with FPN assign anchor boxes of scale within a range to a specific pyramid level. Following this idea, we propose to extract features from FPN according to the downsampling ratio.

Concretely, we assign the images with a downsampling ratio of $2^s$ to $P_{5-s}$ for a 3-level FPN architecture, where we denote the final feature set of FPN as $\{P_3, P_4, P_5\}$ from the finest resolution map to the coarsest resolution map. Similar to RoI pooling operator, we map the window of the component images onto the FPN features and aggregate the features using a global average pooling layer. Subsequently, the dense prediction head processes the features. For the non-FPN framework, we construct a feature pyramid by interpolating the final feature map to the specific sizes.

To encode the augmented views, we use two encoders: the online network $f_\theta$ and the target network $f_{\theta'}$. The target network is implemented as an exponential moving average (EMA) of the online network. The online network is appended with a projector $g_\theta$ and a predictor $h_\theta$, while the target network only has a projector $g_{\theta'}$. We represent each view pair as normalized latent features $\mathbf{u}_{s_i}$ and $\mathbf{v}_{s_i}$, where $\mathbf{I}_{s_i}$ is the image assembled by the subimages whose downsampling ratio is $s_i$,

$$\mathbf{u}_{s_i} = h_\theta \circ g_\theta \circ f_\theta(\mathbf{I}_{s_i}), \qquad \mathbf{v}_{s_i} = g_{\theta'} \circ f_{\theta'}(\mathbf{I}'_{s_i}). \tag{1}$$

We adopt the contrastive loss function in the form of InfoNCE (Van den Oord et al., 2018) to optimize the model:

$$\mathcal{L}_{\mathbf{u}} = -\log \frac{\exp(\mathbf{u} \cdot \mathbf{v}^+/\tau))}{\exp(\mathbf{u} \cdot \mathbf{v}^+/\tau) + \sum_{\mathbf{v}^-} \exp(\mathbf{u} \cdot \mathbf{v}^-/\tau)}, \tag{2}$$

where the subscript of latents are omitted for simplicity. Here, $\mathbf{v}^+$ is the target network's output on the same subimage as $\mathbf{u}$ and the set $\{\mathbf{v}^-\}$ is composed of target network's outputs from other subimages. $\tau$ is a temperature hyper-parameter (Touvron et al., 2021) for $l_2$-normalized latent features. Note that, the feature maps $\mathbf{u}_{s_i}$ and $\mathbf{v}_{s_i}$ are split into vectors based on the coordinates of the subimages. As the number of latent features is sufficiently large, we use the negative samples co-existing in the same batch, following (Bachman et al., 2019; Chen et al., 2020a; Hjelm et al., 2018; Ye et al., 2019). Besides, we adopt a symmetric loss (Caron et al., 2020; Chen & He, 2021; Grill et al., 2020): $\mathcal{L} = \mathcal{L}_{\mathbf{u}} + \mathcal{L}_{\mathbf{v}}$.

### 3.3 MULTI-LEVEL CONTRASTIVE LOSS

To improve the scale consistency, we propose a series of modes to construct the final loss. Specifically, we design four matching strategies for assigning both the positive and the negative samples to the online features. As shown in the right part of Fig. 3, (a) All the images in different sizes target the view in the largest shape, (b) Each image level aims to pull close features from the counterpart level, (c) Latent features match the features from the adjacent levels, and (d) A dense connection is applied to all levels, treating all image resolution equally. The empirical study and comparison are provided in Sec.4.3.

## 4 EXPERIMENTS

In this section, we conduct a comprehensive set of experiments to evaluate our pre-training mechanism on dense prediction tasks, *e.g.*, COCO (Lin et al., 2014) detection, instance segmentation, pose estimation, Cityscapes segmentation (Cordts et al., 2016) and LVIS (Gupta et al., 2019) long tail object detection and segmentation.

### 4.1 PRE-TRAINING SETUP

We pre-train our MCL model on ImageNet-1K (Deng et al., 2009) and COCO (Lin et al., 2014) dataset with LARS (You et al., 2017) optimizer and a batch size of 4096. By default, all the models are pre-trained for 100 epochs on the ImageNet training set, which contains approximately 1.28 million images. The training cost is comparable to supervised pre-training. For pre-training on the non-object-centric dataset, models are optimized for 530 epochs on the COCO training set and unlabeled set (about 241 thousand images) to match the training iteration on ImageNet. We employ the same data augmentation pipeline of BYOL (Grill et al., 2020), which is composed of random crop augmentation, random horizontal flip, color distortion, Gaussian blur, grayscaling and the solarization operation. All the component images are augmented separately with different random seeds but share an augmentation pipeline. The learning rate is linearly warmed up at the first 10 epochs and cosine annealed during the remaining epochs. The learning rate is set based on the batch size: $lr = 1.0 \times BatchSize/256$ and the weight decay is set to $1e^{-5}$. The weights of the target network are updated with a momentum coefficient $m$, starting from 0.99 and increased to 1 in the cosine scheduler same as (Chen et al., 2021; Grill et al., 2020).

Table 1: Comparison with state-of-the-art methods on COCO *val* set. All the models are pre-trained on COCO dataset and finetuned with Mask-RCNN following 1x schedule (Wu et al., 2019).

| Methods | $AP^{bb}$ | $AP^{bb}_{50}$ | $AP^{bb}_{75}$ |
|---|---|---|---|
| Supervised | 38.9 | 59.6 | 42.7 |
| BYOL (Grill et al., 2020) | 39.3(+0.4) | 59.0(-0.6) | 42.8(+0.1) |
| DenseCL (Wang et al., 2021) | 39.8(+0.9) | 59.7(+0.1) | 43.3(+0.6) |
| Self-EMD (Liu et al., 2020) | 40.4(+1.3) | 61.1(+1.5) | 43.7(+1.0) |
| SoCo (Wei et al., 2021) | 40.6(+1.5) | 61.1(+1.5) | 44.4(+1.7) |
| **MCL** | **41.8(+2.9)** | **62.1(+2.5)** | **45.8(+3.1)** |

| Methods | $AP^{mk}$ | $AP^{mk}_{50}$ | $AP^{mk}_{75}$ |
|---|---|---|---|
| Supervised | 35.4 | 56.5 | 38.1 |
| BYOL (Grill et al., 2020) | - | - | - |
| DenseCL (Wang et al., 2021) | 35.8(+0.4) | 56.6(+0.1) | 38.6(+0.5) |
| Self-EMD (Liu et al., 2020) | - | - | - |
| SoCo (Wei et al., 2021) | 36.4(+1.0) | 58.1(+1.6) | 38.1(+0.0) |
| **MCL** | **37.7(+2.3)** | **59.3(+2.8)** | **40.5(+2.4)** |

Table 2: Results on COCO for RetinaNet. All the models are pre-trained on ImageNet and finetuned on COCO with 1x schedule. MCL outperforms all the other state-of-the-art methods.

| Methods | Epoch | $AP^{bb}$ | $AP^{bb}_{50}$ | $AP^{bb}_{75}$ |
|---|---|---|---|---|
| Rand Init | - | 24.5 | 39.0 | 25.7 |
| Supervised | 90 | 37.4 | 56.5 | 39.7 |
| InsDis (Wu et al., 2018) | 200 | 35.5 | 54.1 | 38.2 |
| PIRL (Misra & Maaten, 2020) | 200 | 35.7 | 54.2 | 38.4 |
| MoCo (He et al., 2020) | 200 | 36.3 | 55.0 | 39.0 |
| MoCo v2 (He et al., 2020) | 200 | 37.2 | 56.2 | 39.6 |
| InfoMin (Tian et al., 2020) | 200 | 38.1 | 57.3 | 40.9 |
| SwAV (Caron et al., 2020) | 400 | 36.5 | 56.4 | 38.8 |
| PixPro (Xie et al., 2021b) | 100 | 37.9 | 56.7 | 40.5 |
| SoCo (Wei et al., 2021) | 100 | 38.2 | 57.4 | 40.9 |
| InsLoc (Yang et al., 2021) | 200 | 36.4 | 55.3 | 39.0 |
| DenseCL (Wang et al., 2021) | 200 | 37.6 | 56.3 | 40.3 |
| **MCL** | **100** | **39.1** | **58.5** | **41.8** |

Table 3: Comparison with SOTA methods on COCO by using Mask R-CNN. All the detectors are evaluated on COCO *val* 2017 set. "-" means that the results are missing in the source paper. MCL outperforms all the other SOTA SSL methods while significantly reducing the training epochs.

| Methods | Epoch | 1× Schedule | | | | | | 2× Schedule | | | | | |
|---|---|---|---|---|---|---|---|---|---|---|---|---|---|
| | | $AP^{bb}$ | $AP^{bb}_{50}$ | $AP^{bb}_{75}$ | $AP^{mk}$ | $AP^{mk}_{50}$ | $AP^{mk}_{75}$ | $AP^{bb}$ | $AP^{bb}_{50}$ | $AP^{bb}_{75}$ | $AP^{mk}$ | $AP^{mk}_{50}$ | $AP^{mk}_{75}$ |
| Rand Init | - | 31.0 | 49.5 | 33.2 | 28.5 | 46.8 | 30.4 | 38.4 | 57.5 | 42.0 | 34.7 | 54.8 | 37.2 |
| Supervised | 90 | 38.9 | 59.6 | 42.7 | 35.4 | 56.5 | 38.1 | 41.3 | 61.3 | 45.0 | 37.3 | 58.3 | 40.3 |
| Briging(Liu et al., 2022) | 100 | 36.5 | - | - | 33.6 | - | - | - | - | - | - | - | - |
| MoCo (He et al., 2020) | 200 | 38.5 | 58.9 | 42.0 | 35.1 | 55.9 | 37.7 | 40.8 | 61.6 | 44.7 | 36.9 | 58.4 | 39.7 |
| MoCo v2 (Chen et al., 2020c) | 200 | 40.4 | 60.2 | 44.2 | 36.4 | 57.2 | 38.9 | 41.7 | 61.6 | 45.6 | 37.6 | 58.7 | 40.5 |
| MoCo v3 (Chen et al., 2021) | 300 | 41.1 | 61.6 | 45.0 | 37.3 | 58.8 | 40.0 | - | - | - | - | - | - |
| InfoMin (Tian et al., 2020) | 200 | 40.6 | 60.6 | 44.6 | 36.7 | 57.7 | 39.4 | 42.5 | 62.7 | 46.8 | 38.4 | 59.7 | 41.4 |
| UniVIP (Li et al., 2022) | 200 | 41.6 | - | - | 37.6 | - | - | 43.1 | - | - | 38.8 | - | - |
| BYOL (Grill et al., 2020) | 300 | 40.4 | 61.6 | 44.1 | 37.2 | 58.8 | 39.8 | 42.3 | 62.6 | 46.2 | 38.3 | 59.6 | 41.1 |
| SwAV (Caron et al., 2020) | 400 | 41.2 | 62.1 | 45.0 | 37.3 | 59.0 | 40.2 | 42.3 | 62.8 | 46.3 | 38.2 | 60.0 | 41.0 |
| VICRegL(Bardes et al., 2022) | 400 | 37.3 | - | - | 34.1 | - | - | - | - | - | - | - | - |
| DINO (Caron et al., 2021) | 800 | 41.2 | - | - | 37.1 | - | - | 42.3 | - | - | 38.1 | - | - |
| UniGrad (Tao et al., 2022) | 800 | 42.0 | 62.6 | 45.7 | 37.9 | 59.7 | 40.7 | - | - | - | - | - | - |
| SparK (Tian et al., 2023) | 1600 | 41.6 | - | - | 37.7 | - | - | - | - | - | - | - | - |
| PLRC(Bai et al., 2022) | 100 | 39.3 | 58.8 | 43.1 | 35.5 | 55.9 | 37.9 | - | - | - | - | - | - |
| SoCo(Wei et al., 2021) | 100 | 42.3 | 62.5 | 46.5 | 37.6 | 59.1 | 40.5 | 43.2 | 63.3 | 47.3 | 38.8 | 60.6 | 41.9 |
| DCL (Wang et al., 2021) | 200 | 40.3 | 59.9 | 44.3 | 36.4 | 57.0 | 39.2 | 41.2 | 61.9 | 45.1 | 37.3 | 58.9 | 40.1 |
| ReSim (Xiao et al., 2021) | 200 | 39.8 | 60.2 | 43.5 | 36.0 | 57.1 | 38.6 | 41.4 | 61.9 | 45.4 | 37.5 | 59.1 | 40.3 |
| RCL (Xu et al., 2022) | 200 | 40.4 | 61.3 | 44.2 | 36.7 | 58.2 | 39.4 | 42.1 | 62.9 | 45.9 | 38.0 | 60.0 | 40.7 |
| DetCon (Hénaff et al., 2021) | 300 | 42.0 | - | - | 37.8 | - | - | - | - | - | - | - | - |
| PixPro (Xie et al., 2021b) | 400 | 41.4 | 61.6 | 45.4 | - | - | - | - | - | - | - | - | - |
| InsLoc (Yang et al., 2021) | 400 | 42.0 | 62.3 | 45.8 | 37.6 | 59.0 | 40.5 | 43.3 | 63.6 | 47.3 | 38.8 | 60.9 | 41.7 |
| SCRL (Roh et al., 2021) | 800 | 41.3 | 62.4 | 45.0 | 37.7 | 59.6 | 40.7 | - | - | - | - | - | - |
| MCL | 100 | 42.5 | 62.8 | 46.9 | 38.2 | 59.8 | 41.2 | 43.4 | 63.6 | 47.5 | 39.1 | 60.8 | 41.9 |
| **MCL** | **400** | **43.0** | **63.3** | **47.4** | **38.6** | **60.2** | **41.7** | **44.0** | **64.2** | **48.3** | **39.5** | **61.3** | **42.7** |

## 4.2 RESULTS ON DOWNSTREAM TASKS

**Pre-training on Non-object-centric Dataset.** ImageNet is an object-centric dataset, which introduces dataset biases into pre-training and costs more effort to collect than non-iconic images. As indicated in Tab. 1, MCL still obtains a large improvement, 1.4 $AP^{bb}$/1.3 $AP^{mk}$ over SoCo (Wei et al., 2021). This result demonstrates that MCL is robust to dataset and benefits mainly from the scale-invariance and precise localization representation rather than the dataset bias. Besides, the results manifest that pixel-level SSL methods and object-level methods fail in the non-iconic scenario.

**COCO Object Detection and Instance Segmentation.** Object detection and instance segmentation require simultaneous object location and classification while handling large variance of object size. We adopt Mask-RCNN (He et al., 2017) and RetinaNet (Lin et al., 2017) with ResNet-50 FPN backbone as detectors to evaluate the models pre-trained on ImageNet and COCO dataset. As shown in Tab. 3, MCL outperforms the state-of-the-art (SOTA) unsupervised pre-training methods on the COCO 1x and 2x schedules with only 100 training epochs, achieving 42.5 $AP^{bb}$/38.2 $AP^{mk}$ and 43.4 $AP^{bb}$/39.1 $AP^{mk}$ on the 1x and 2x schedule, respectively. Our method surpasses the supervised counterpart by 3.6 $AP^{bb}$ and 2.8 $AP^{mk}$ on 1x schedule, showing that MCL accelerates the model converging on the downstream tasks.

Table 4: Results of finetuning model in a low data regime. One-stage detection fine-tuned on COCO 1%, 2%, 5% and 10% data. All methods **except** MCL are pre-trained 200 epochs on ImageNet. MCL is per-trained for 100 epochs.

| Methods | 1% Data | | | 2% Data | | | 5% Data | | | 10% Data | | |
|---|---|---|---|---|---|---|---|---|---|---|---|---|
| | AP | $AP_{50}$ | $AP_{75}$ | AP | $AP_{50}$ | $AP_{75}$ | AP | $AP_{50}$ | $AP_{75}$ | AP | $AP_{50}$ | $AP_{75}$ |
| Rand Init | 1.4 | 3.5 | 1.0 | 2.5 | 5.6 | 2.0 | 3.6 | 7.4 | 3.0 | 3.7 | 7.5 | 3.2 |
| Supervised | 8.2 | 16.2 | 7.2 | 11.2 | 21.7 | 10.3 | 16.5 | 30.3 | 15.9 | 19.6 | 34.5 | 19.7 |
| MoCo(He et al., 2020) | 7.0(-1.2) | 13.5(-2.7) | 6.5(-0.7) | 10.3(-0.9) | 19.2(-2.5) | 9.7(-0.6) | 15.0(-1.5) | 27.0(-3.3) | 14.9(-1.0) | 18.2(-1.4) | 31.6(-2.9) | 18.4(-1.3) |
| MoCo v2(Chen et al., 2020c) | 8.4(+0.2) | 15.8(-0.4) | 8.0(+0.8) | 12.0(+0.8) | 21.8(+0.1) | 11.5(+1.2) | 16.8(+0.3) | 29.6(-0.7) | 16.8(+0.9) | 20.0(+0.4) | 34.3(-0.2) | 20.2(+0.5) |
| DetCo(Xie et al., 2021a) | 9.9(+1.7) | 19.3(+3.1) | 9.1(+1.9) | 13.5(+2.3) | 25.1(+3.4) | 12.7(+2.4) | 18.7(+2.2) | 32.9(+2.6) | 18.7(+2.8) | 21.9(+2.3) | 37.6(+3.1) | 22.3(+2.6) |
| **MCL** | **12.1**(+3.9) | **22.6**(+6.4) | **11.6**(+4.4) | **15.4**(+4.2) | **27.0**(+5.3) | **15.6**(+5.3) | **20.7**(+4.2) | **35.5**(+5.2) | **20.7**(+4.8) | **23.8**(+4.2) | **39.6**(+5.1) | **24.2**(+4.5) |

Table 5: Transfer Learning on LVIS dataset using Mask R-CNN with R50-FPN trained for $180k$ iterations. MCL significantly improves the performance on **rare** categories by 4.3 $AP^{bb}$/4.3 $AP^{mk}$.

| Methods | $AP^{bb}$ | $AP^{bb}_r$ | $AP^{bb}_c$ | $AP^{bb}_f$ | $AP^{mk}$ | $AP^{mk}_r$ | $AP^{mk}_c$ | $AP^{mk}_f$ |
|---|---|---|---|---|---|---|---|---|
| Supervised | 23.9 | 10.2 | 21.8 | 32.2 | 23.1 | 11.1 | 21.6 | 30.1 |
| SoCo | 24.3 | 12.2 | 21.7 | 32.4 | 23.5 | 13.7 | 21.4 | 30.2 |
| **MCL** | **26.2** | **14.5** | **23.4** | **34.0** | **25.5** | **15.4** | **23.9** | **31.6** |

Table 6: Results of Mask R-CNN on COCO Keypoint dataset. The results demonstrate that MCL is available for other dense prediction task besides detection task.

| Methods | $AP^{bb}$ | $AP^{bb}_{50}$ | $AP^{bb}_{75}$ | $AP^{kp}$ | $AP^{kp}_{50}$ | $AP^{kp}_{75}$ |
|---|---|---|---|---|---|---|
| Supervised | 57.5 | 84.0 | 63.0 | 65.6 | 87.0 | 71.3 |
| **MCL** | **58.9**(+1.4) | **85.2**(+1.2) | **65.1**(+2.1) | **66.8**(+1.2) | **87.8**(+0.8) | **72.8**(+1.5) |

*To verify the extendability of MCL, we conduct experiments on RetinaNet* (Lin et al., 2017), which is different from the pre-training architecture. We follow the standard COCO 1x schedule and include SyncBN in the backbone and FPN for a fair comparison. Tab. 2 shows that MCL exceeds the supervised baseline by 1.7 $AP^{bb}$.

**Finetune in a Low Data Regime.** One of the purposes of pre-training is to improve the target task performance in a low data regime. Therefore, we conduct experiments on a mini version of the COCO dataset. Specifically, we randomly sample 1, 2, 5 and 10% of COCO training data as the labeled dataset. To avoid overfitting, we finetune the detectors with 12k iterations. Other settings are the same as COCO 1x schedule. Tab. 4 indicates that MCL has strong generalization than other methods and outperforms the supervised counterparts by about 4 AP. This result shows that MCL can be extended to semi-supervised learning for object detection as a consistency regularization.

**Finetune on LVIS Dataset.** Compared with COCO, LVIS v1 dataset (Gupta et al., 2019) is more challenging due to the long tail distribution, which contains 1203 categories. To demonstrate the effectiveness and generality of our method, we finetune a Mask R-CNN model and follow the standard LVIS v1 1x training schedule. Tab. 5 shows that MCL significantly improves the performance on rare categories by 4.3 $AP^{bb}$/4.3 $AP^{mk}$, which is much larger than the improvement on the common and frequent categories.

**ADE20K, Cityscapes and COCO KeyPoint Dataset.** To evaluate our method on other downstream tasks, we choose ADE20K, Cityscapes and COCO Keypoint dataset. We evaluate MCL with Mask R-CNN and Semantic FPN on Cityscapes dataset, with UperNet on ADE20K-847 datasets. For the COCO Keypoint dataset, we attach the standard keypoint head on Mask R-CNN. As shown in Tab. 6 and Tab. 9, MCL achieves 35.7 $AP^{mk}$ on Cityscapes instance segmentation task, 76.1 mIoU on the semantic segmentation task and 66.8 $AP^{kp}$ on COCO Keypoint task. The superior results show that MCL is suitable for dense prediction tasks besides the detection task.

**Supervised Pre-training on Transformer and CNN with MCL Pretext Task.** It seems like a foregone conclusion that self-supervised pre-training surpasses the supervised counterpart on downstream tasks. However, we find that MCL pretext task facilitates the finetuning of supervised pre-training. For a fair comparison, we pre-train models with the same epochs as the normal supervised learning. Concretely, ResNet-50 (He et al., 2016) is trained with 100 epochs and Swin-T (Liu et al., 2021) is trained with 300 epochs. The other hyperparameters keep unchanged. The evaluation is still based on COCO 1x training schedule. Mask R-CNN with Swin-T is finetuned with MMDetection (Chen et al., 2019). Tab. 7 shows that MCL outperforms 3.2 $AP^{bb}$/2.4 $AP^{mk}$ over supervised counterpart for ResNet-50 and the unsupervised counterpart as shown in Tab. 3. The results demonstrate that *MCL pretext task is effective for the Transformer architecture with attention module*, surpassing the baseline by 0.8 $AP^{bb}$/0.7 $AP^{mk}$.

**Comparison with other augmentation.** Mosaic (Zhang et al., 2021) and Copy-Paste (Ghiasi et al., 2021) are similar to MCL. Compared with these methods, MCL contains sub-images in multi-sizes, which improves scale consistency, and targets the view in the largest resolution, learning more se-

Table 7: Results on COCO with supervised pretraining. Models are pre-trained with the same data augmentation as standard supervised pre-training (Vanilla).

Table 8: Results of Mask-RCNN under 1x schedule on COCO *val* set. We conduct experiments with Mosiac and Copy-Paste data augmentation under both supervised and self-supervised pre-training settings. The results demonstrate that MCL outperform the other methods.

Table 9: Results of Mask R-CNN (Instance Segmentation) and Semantic FPN on Cityscapes dataset. Results of UperNet on ADE20K-847 datasets (Semantic Segmentation).

| Methods | ResNet-50 | | Swin-T | |
|---|---|---|---|---|
| | $AP^{bb}$ | $AP^{mk}$ | $AP^{bb}$ | $AP^{mk}$ |
| Vanilla | 38.9 | 35.4 | 43.9 | 39.6 |
| **MCL** | **42.1** | **37.8** | **44.7** | **40.3** |

| Methods | Sup | Sup + Mosiac | SSL + Mosiac | SSL + Copy-Paste | MCL |
|---|---|---|---|---|---|
| $AP^{bb}$ | 38.9 | 40.1 | 41.1 | 42.0 | **43.0** |
| $AP^{mk}$ | 35.4 | 36.3 | 37.2 | 37.6 | **38.6** |

| Methods | Cityscapes | | | ADE20k-847 |
|---|---|---|---|---|
| | mIoU | $AP^{mk}$ | $AP^{mk}_{50}$ | mIoU |
| Supervised | 72.9 | 31.8 | 58.5 | 17.2 |
| **MCL** | **76.1** | **35.7** | **63.9** | **20.8** |

Table 10: Ablation studies on COCO for the proposed MCL method. All the models are pretrained on ImageNet dataset for 100 epochs and finetuned on COCO dataset following 1x schedule Wu et al. (2019). In (a), we set SwAV Carion et al. (2020) with Multi-Crop as baseline. The loss indicators in (b) are same as those in Fig. 3. None in (c) indicates that the subimages are not smoothed. Div in (d) means all the non-normalization layer parameters are divided by a fixed number. In (e), B means that only the backbone is pre-trained, F indicates FPN neck and H is the detection head. The results are reported with Mask R-CNN in all tables except (f), in which the results of RetinaNet are provided.

| Level | $AP^{bb}$ | $AP^{bb}_{50}$ | $AP^{bb}_{75}$ | $AP^{mk}$ | $AP^{mk}_{50}$ | $AP^{mk}_{75}$ |
|---|---|---|---|---|---|---|
| SwAV | 40.7 | 60.9 | 44.6 | 36.8 | 58.0 | 39.8 |
| 2 | 41.4 | 61.6 | 45.4 | 37.3 | 58.6 | 39.9 |
| 3 | 41.8 | 61.7 | 45.5 | 37.6 | 58.8 | 40.2 |
| 4 | 42.5 | 62.8 | 46.9 | 38.2 | 59.8 | 41.2 |

(a) Study on Downsampling Level.

| Weight Decay | $AP^{bb}$ | $AP^{bb}_{50}$ | $AP^{bb}_{75}$ | $AP^{mk}$ | $AP^{mk}_{50}$ | $AP^{mk}_{75}$ |
|---|---|---|---|---|---|---|
| 1.5e-6 | 41.7 | 62.2 | 45.9 | 37.8 | 59.3 | 40.6 |
| 1.5e-6 Div 1.5 | 41.8 | 62.3 | 45.9 | 37.8 | 59.3 | 40.6 |
| 1.5e-6 Div 2 | 42.2 | 62.4 | 46.4 | 37.8 | 59.4 | 40.6 |
| 5e-6 | 42.3 | 62.6 | 46.7 | 38.0 | 59.7 | 40.8 |
| 1e-5 | 42.5 | 62.8 | 46.9 | 38.2 | 59.8 | 41.2 |

(d) Study on Weight Decay.

| Loss | $AP^{bb}$ | $AP^{bb}_{50}$ | $AP^{bb}_{75}$ | $AP^{mk}$ | $AP^{mk}_{50}$ | $AP^{mk}_{75}$ |
|---|---|---|---|---|---|---|
| a | 42.5 | 62.8 | 46.9 | 38.2 | 59.8 | 41.2 |
| b | 41.8 | 61.8 | 45.7 | 37.4 | 58.8 | 40.2 |
| c | 42.0 | 62.2 | 46.0 | 37.8 | 59.2 | 40.8 |
| d | 41.0 | 61.5 | 44.6 | 37.1 | 58.6 | 40.0 |

(b) Study on Multi-Level Loss.

| Arch. | $AP^{bb}$ | $AP^{bb}_{50}$ | $AP^{bb}_{75}$ | $AP^{mk}$ | $AP^{mk}_{50}$ | $AP^{mk}_{75}$ |
|---|---|---|---|---|---|---|
| B | 41.1 | 61.3 | 45.4 | 37.2 | 58.6 | 39.9 |
| B+F | 41.5 | 61.6 | 45.5 | 37.4 | 58.6 | 40.1 |
| B+F+H | 42.5 | 62.8 | 46.9 | 38.2 | 59.8 | 41.2 |

(e) Study on Architecture Alignment.

| Std | $AP^{bb}$ | $AP^{bb}_{50}$ | $AP^{bb}_{75}$ | $AP^{mk}$ | $AP^{mk}_{50}$ | $AP^{mk}_{75}$ |
|---|---|---|---|---|---|---|
| None | 42.5 | 62.8 | 46.9 | 38.2 | 59.8 | 41.2 |
| 0.75 | 42.2 | 62.3 | 46.3 | 37.8 | 59.4 | 40.7 |
| 0.5 | 42.0 | 62.3 | 45.8 | 37.8 | 59.3 | 40.6 |

(c) Study on Boundary Smoothness.

| Epoch | $AP^{bb}$ | $AP^{bb}_{50}$ | $AP^{bb}_{75}$ | $AP^{bb}_{s}$ | $AP^{bb}_{m}$ | $AP^{bb}_{l}$ |
|---|---|---|---|---|---|---|
| 100 | 39.1 | 58.5 | 41.8 | 26.5 | 43.7 | 47.3 |
| 200 | 39.5 | 59.2 | 42.7 | 25.8 | 44.2 | 47.9 |
| 400 | 39.9 | 59.8 | 42.7 | 26.7 | 44.4 | 48.3 |

(f) Study on Training Epoch.

mantic information. As shown in Tab.8, MCL outperforms supervised and self-supervised Mosiac pre-training.

## 4.3 ABLATION STUDY

**Montage Level.** Our method is based on montage assembly over the multi-level downsampled images and contrastive learning. We investigate the effect of the montage level by pretraining models with different downsampling levels. By comparing the penultimate and the last line in Tab. 10a, the results show that the representation of fine-grained objects, whose size is less than $28 \times 28$ pixels, is important for the object detector. This phenomenon demonstrates that the alignment of instance scale range is important between the pre-training dataset and the target dataset and that object detectors benefit from our multi-level scale consistency constrastive learning.

**Multi-Level Contrastive Loss.** As shown in Fig. 3, we propose a series of meaningful positive pair matching strategies. The loss indicators in Tab. 10b are the same as those in Fig. 3. We find that setting the images with the largest resolution as positive pair samples leads to the best result. This result is reasonable because a higher resolution typically yields a better representation. The reason why $b$ loss mode is inferior can be that the supervision from the counterpart level lacks semantic information for the small component images. The result of $c$ loss mode is slightly better than $b$ mode due to the feature matching across levels. The cause of $d$ loss mode yielding a lower result

than $c$ is probably that the representation from the low-resolution image is inferior to that from the high-resolution image.

**Boundary smoothness.** The boundary of the subimages is a sudden appearance change, which may be problematic for self-supervised representation learning. To verify if the boundary is problematic, we smooth the boundary with a Gaussian mask. The standard deviation of the mask $(\sigma_x, \sigma_y)$ is set to $(k \times W, k \times H)$, where $k$ is set to 0.5 or 0.75, H is the height of the subimage and W is the width of the subimage. The Gaussian mask softens the boundary gap and highlights the center. As indicated in Tab. 10c, assembling the origin subimages directly yields the best result. We believe that the Average RoI-pooling alleviates the boundary issue and the Gaussian mask may cause the information loss of some objects located at the boundary.

**Weight Decay.** Weight norm is an important factor in the alignment between pre-training and fine-tuning. We empirically demonstrate this by modifying the weight decay, which influences the weight norm of the converged model. Typically, the weight decay is set to $1e^{-6}$ for LARS optimizer, which is widely adopted in unsupervised pre-training works (Caron et al., 2020; Chen et al., 2020a; 2021; Grill et al., 2020). We set the weight decay from $1.5e^{-6}$ to $1e^{-5}$ to evaluate the effect. The results in Tab. 10d show that a large weight decay leads to a superior result. Additionally, we divide the non-normalization layer parameters by a fixed number to downscale the weight norm. The results show that reducing the weight norm of a model pre-trained with a small weight decay leads to a non-negligible improvement. Normalization techniques exist in many mainstream models (Dosovitskiy et al., 2020; He et al., 2016; Huang et al., 2017; Liu et al., 2021; Tolstikhin et al., 2021; Sandler et al., 2018; Wu & He, 2018), which makes output resilient to the parameter scale, we take Batch Normalization (Ioffe & Szegedy, 2015) for example: $\mathrm{BN}(Wx) = \mathrm{BN}((\alpha W)x)$, and we can show that: $\frac{\partial \mathrm{BN}((\alpha W)x)}{\partial \alpha W} = \frac{1}{\alpha} \cdot \frac{\partial \mathrm{BN}(Wx)}{\partial W}$, where $\alpha$ is a positive scalar. In the case that $\alpha < 1$, the gradient of parameter $W$ is magnified. Following the SGD update rule, the model weight of $t + 1$ step is $W_{t+1} = W_t - \eta \frac{1}{\alpha} \frac{\partial \mathrm{BN}(Wx)}{\partial W}$, where $\eta$ is the learning rate. Suppose that **the learning rate is suitable** and **the weight is well-initialized**, the relatively small weight norm leads to a faster convergence, compared with the large model weight.

**Architecture Alignment.** We ablate each architecture component step by step to verify the importance of alignment of the downstream and pre-train model architecture. The pre-trained weights of FPN neck and Detection Head are ablated. Tab. 10e reports the studies, in which the baseline achieves 41.1 $AP^{bb}$/37.2 $AP^{mk}$. FPN neck further improves the performance to 41.5 $AP^{bb}$/37.4 $AP^{mk}$ and Detection Head finally improves the result to 42.5 $AP^{bb}$/38.2 $AP^{mk}$. Pre-trained Detection Head leads to additional gain for Mask R-CNN, while MCL also outperforms other state-of-the-art methods on the RetinaNet model (as shown in Tab. 2), which has a different detection head and FPN architecture from Mask R-CNN. This phenomenon demonstrates that MCL is somewhat robust to model architecture.

**Training Epochs.** We extend the training epochs to 200 epochs and 400 epochs on the ImageNet. The pre-trained model is finetuned using RetinaNet with the standard 1x COCO schedule. Pre-trained for 200 epochs, MCL improves the detection result to 39.5 AP. Another 200 training epochs increase the performance by 0.4 AP, which means that long training schedule improves the performance.

## 5 CONCLUSION

In this work, we introduce a novel self-supervised framework based on multi-level contrastive learning. Our method learns regional representation for precise localization, scale consistency among multi-scale crops, and semantic representations that generalize well on the dense prediction tasks. The montage assembly explicitly encodes the absolute position and scale information. Multi-level contrastive learning aligns the feature map with the image region and regularizes the scale consistency. Besides, we empirically explore the alignment between pre-training and finetuning by investigating the transfer performance of applying the proposed pretext task in the supervised learning scenario. Our experiment results demonstrate the state-of-the-art transfer performance on various dense prediction tasks, while significantly reducing the pre-training epochs. A further fine-grain representation learning under our framework may lead to a promising result.

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
