

Figure 4: Overview of MCL applied in the supervised scenario. All the input images are augmented via the same pipeline with different seeds. The input images are downsampled and assembled at different levels. The ground truths are assembled in the same order as the montage images. The multi-level cross-entropy loss is applied to optimize the model.

# MULTI-LEVEL CONTRASTIVE LEARNING FOR DENSE PREDICTION TASK

**Anonymous authors**

## A APPENDIX

### A.1 ADDITIONAL IMPLEMENTATION DETAILS

**Multi-Level Contrastive Loss.** As described in Sec. 3.1, the images are downsampled according to the level index and assembled in a montage manner. The numbers of montage image in level $i$ is $\frac{B}{4^i}$, where $i$ starts from 0 to $S - 1$ and $B$ is the batch size. Therefore, the montage assembly increases the computational cost marginally and the upper bounder is twice the baseline batch size. Since the highest resolution images typically yield the best semantic representation and the empirical result in Sec. 4.3, the first level contrastive loss (on the highest resolution images) is important to the representation learning. As a consequence, we assign different loss weight to each level, $\frac{1}{2^{(i+1)}}$ for the $i$-th level. The detection head is a shared 4 *CONV* head without *fc* layer across levels, which are not loaded in the RetinaNet detector. We attach a global average pooling layer on the detection head to aggregate the features because averaging implicitly encourages a high response region. Both the projection head and prediction head are 2-layer MLPs whose hidden layer dimension is 2048. The final linear layer has a 256-dimension output and a final BN layer is attached to the projection head to accelerate the convergence.

**Multi-Level Supervised Learning.** We extend MCL to the supervised learning scenario to demonstrate the importance of the alignment between the pretext task and downstream tasks. As illustrated in Fig. 4, we generate $S$ augmented views for the model, which are downsampled to $\frac{1}{2^s}$ original size. We set $s$ as the level index, which starts from 0 to $S - 1$. Different from self-supervised learning, we simply adopt the same optimizer as the normal setting, SGD optimizer for ResNet and AdamW optimizer for Swin-Transformer. The other hyperparameters are the same as the baseline counterparts.

## B ADDITIONAL RESULTS

**Long Finetuning Schedule.** The experiments in Sec. 4 mainly follow the 1x and 2x schedule, which are not long enough for detectors to be fully converged. We extend the training schedule to 6x schedule, *i.e.* 540$k$ iterations. Tab. 11 shows that MCL pre-trained with 400 epochs achieves 41.2

Table 11: Results of the long training schedule for RetinaNet finetuned on COCO with 90$k$, 180$k$, and 540$k$. MCL not only accelerates the convergence but also improves the final performance.

| Methods | Epoch | 1x schedule | | | 2x schedule | | | 6x schedule | | |
|---|---|---|---|---|---|---|---|---|---|---|
| | | AP | $AP_{50}$ | $AP_{75}$ | AP | $AP_{50}$ | $AP_{75}$ | AP | $AP_{50}$ | $AP_{75}$ |
| Supervised | 90 | 37.4 | 56.6 | 39.7 | 38.8 | 58.7 | 41.2 | 39.2 | 58.6 | 42.1 |
| MoCo v2 | 800 | 37.9(+0.5) | 57.1(+0.5) | 40.4(+0.7) | 39.8(+1.0) | 59.3(+0.6) | 42.8(+1.6) | 40.2(+1.0) | 59.9(+1.3) | 43.1(+1.0) |
| **MCL** | 400 | **39.9**(+2.5) | **59.8**(+3.2) | **42.7**(+3.0) | **41.2**(+2.4) | **61.1**(+2.4) | **44.0**(+2.8) | **41.4**(+2.2) | **61.1**(+2.5) | **44.5**(+2.4) |

Table 12: Comparison with SOTA methods on COCO by using Mask R-CNN with R50-C4. All the detectors are evaluated on COCO *val* 2017 set. "-" means that the results are missing in the source paper. MCL achieves SOTA results while significantly reducing the training epochs.

| Methods | Epoch | $1\times$ Schedule | | | | | | $2\times$ Schedule | | | | | |
|---|---|---|---|---|---|---|---|---|---|---|---|---|---|
| | | $AP^{bb}$ | $AP^{bb}_{50}$ | $AP^{bb}_{75}$ | $AP^{mk}$ | $AP^{mk}_{50}$ | $AP^{mk}_{75}$ | $AP^{bb}$ | $AP^{bb}_{50}$ | $AP^{bb}_{75}$ | $AP^{mk}$ | $AP^{mk}_{50}$ | $AP^{mk}_{75}$ |
| Rand Init | - | 26.4 | 44.0 | 27.8 | 29.3 | 46.9 | 30.8 | 35.6 | 54.6 | 38.2 | 31.4 | 51.5 | 33.5 |
| Supervised | 90 | 38.2 | 58.2 | 41.2 | 33.3 | 54.7 | 35.2 | 40.0 | 59.9 | 43.1 | 34.7 | 56.5 | 36.9 |
| MoCo (He et al., 2020) | 200 | 38.5 | 58.3 | 41.6 | 33.6 | 54.8 | 35.6 | 40.7 | 60.5 | 44.1 | 35.4 | 57.3 | 37.6 |
| SimCLR (Chen et al., 2020a) | 200 | - | - | - | - | - | - | 39.6 | 59.1 | 42.9 | 34.6 | 55.9 | 37.1 |
| MoCo v2 (Chen et al., 2020b) | 800 | 39.3 | 58.9 | 42.5 | 34.3 | 55.7 | 36.5 | 41.2 | 60.9 | 44.6 | 35.8 | 57.7 | 38.2 |
| InfoMin (Tian et al., 2020) | 200 | 39.0 | 58.5 | 42.0 | 34.1 | 55.2 | 36.3 | 41.3 | 61.2 | 45.0 | 36.0 | 57.9 | 38.3 |
| BYOL (Grill et al., 2020) | 300 | - | - | - | - | - | - | 40.3 | 60.5 | 43.9 | 35.1 | 56.8 | 37.3 |
| SwAV (Caron et al., 2020) | 400 | - | - | - | - | - | - | 39.6 | 60.1 | 42.9 | 34.7 | 56.6 | 36.6 |
| SimSiam (Chen & He, 2021) | 200 | 39.2 | 59.3 | 42.1 | 34.4 | 56.0 | 36.7 | - | - | - | - | - | - |
| PixPro (Xie et al., 2021) | 400 | 40.5 | 59.8 | 44.0 | - | - | - | - | - | - | - | - | - |
| SoCo (Wei et al., 2021) | 100 | 40.4 | 60.4 | 43.7 | 34.9 | 56.8 | 37.0 | 41.1 | 61.0 | 44.4 | 35.6 | 57.5 | 38.0 |
| MCL | 100 | 40.0 | 60.3 | 43.2 | 34.7 | 56.7 | 36.7 | **41.7** | **61.7** | **45.4** | **36.1** | **58.1** | **38.5** |

$AP^{bb}$ as 2x schedule is applied. MCL with 6x schedule still surpasses the supervised counterpart and MoCo v2 pre-trained with 800 epochs. These results prove that pre-training not only accelerates the convergence but also improves the final performance.

**Mask R-CNN with C4 on COCO.** As described in Sec. 3.1, MCL is compatible with the non-FPN framework. We construct a feature pyramid by directly interpolating the single-level feature in bilinear mode to the specific sizes. The results in Tab. 12 show that MCL achieves SOTA results while significantly reducing the training epochs. Our method achieves a superior result with a 1x schedule and benefits from a long finetune schedule, *i.e.* 2x COCO schedule. We believe that the reason that MCL yields an inferior result on Mask R-CNN C4, compared with Mask R-CNN FPN, is that Mask R-CNN C4 has only a single-level feature for detection, which yields a lower baseline on the small object detection.

**Linear Evaluation on ImageNet-1K.** MCL learns global semantic representation besides scale consistency and regional localization. We present the ImageNet-1K linear evaluation results for reference. Following the common setting (Caron et al., 2020; Grill et al., 2020; He et al., 2020), data augmentation contains random crop with resize of $224 \times 224$ pixels and random flip. Only the backbone network parameters are loaded and frozen. The classification head is trained for 100 epochs, using an SGD optimizer with a momentum of 0.9 and a batch size of 256. The learning rate starts with 10 and the weight decay is 0. In the test phase, the data augmentation is a center crop from a resized $256 \times 256$ image. Tab. 13 shows that MCL surpasses SoCo on ImageNet linear evaluation, learning a semantic global representation. Compared with the self-supervised learning methods for image classification, MCL outperforms them on dense prediction tasks, while underperforms some of them on the linear evaluation. This phenomenon shows that the improvement on upstream task does not guarantee a better transfer performance on downstream tasks, due to the task misalignment.

## B.1 ANALYSIS

As discussed in Sec. 1, the scale of objects varies in a small range for ImageNet classification model, whereas the scale deviation of MS-COCO dataset (Lin et al., 2014) is large across object instances for detectors. As shown in Fig. 5, the standard deviation of the scale of instances in MS-COCO is 188.4, while that of ImageNet is 56.7. Our MCL assembles multi-scale images to augment the scale range and distribution.

Besidse, MCL encodes the location of each sub-image explicitly to learn representations for the dense prediction tasks. Fig. 6 shows that MCL performs better than baseline at a high IoU threshold

| Methods | 100 ep | 400 ep |
|---|---|---|
| Supervised | 76.5 | - |
| SoCo | 59.7 | 62.6 |
| SimCLR | 66.5 | 69.8 |
| MoCo v2 | 67.4 | 71.0 |
| SwAV | 66.5 | 70.7 |
| SimSiam | 68.1 | 70.8 |
| **MCL** | **69.9** | **71.5** |

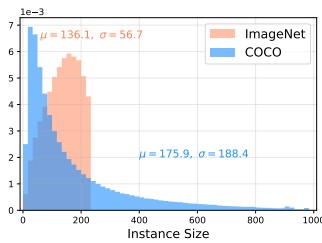

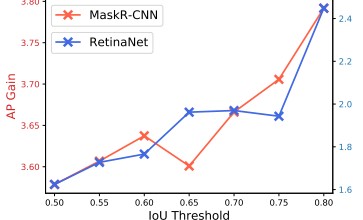

Table 13: Comparison with state-of-the-art self-supervised learning methods on ImageNet-1K **linear evaluation** with the ResNet-50 backbone.

Figure 5: Instance Size Distribution. For the COCO dataset, all the images are resized to (1333, 800) shape. For the ImageNet dataset, all the images are resized to $224 \times 224$ to calculate the statistics.

Figure 6: The $AP^{bb}$ gain of MCL over the supervised counterpart. MCL performs better than the baseline at a higher IoU threshold, indicating that the MCL features have better localization capability.

for both RetinaNet and Mask-RCNN detectors, which demonstrates that pre-training is beyond a strong regularization technique. MCL successfully transfers the strong prior knowledge for precise localization from the pretext task to the downstream tasks.

**Is Montage assembly the same as multi-crop in SwAV?** Montage assembly is beyond multi-crop. CNNs are **not** scale-invariant (Guo et al., 2022) or **not** shift-invariant (Zhang, 2019). CNN delivers **absolute** position information for an image by zero-padding (Islam et al., 2020). Therefore, the feature of a montage images is not the same as assembling the features of each component image, $f([x_1, x_2; x_3, x_4]) \neq [f(x_1), f(x_2); f(x_3), f(x_4)]$. Our pretext task aims to improve **scale consistency** and **shift invariance**, which are **crucial** for dense prediction tasks. Check **Tab**.10(c) for the study of boundary smoothness.

**Training cost.** The batch size for each level is B, B/4, B/16. That results in a training cost is about 1.3125 times that of standard training for the online network. This cost is considered **cost-effective** for the improvement achieved. SoCo constructs 3 views, two 224x224 and one 112x112, resulting in a training cost of 1.25 times that of standard training for online network.