# OpenReview forum: "Multi-Level Contrastive Learning for Dense Prediction Task"
_ICLR.cc/2024/Conference — ICLR 2024 Conference Withdrawn Submission_

### Official Review · Reviewer_dPd7 · 2023-10-19

**Soundness:** 2 fair
**Presentation:** 2 fair
**Contribution:** 2 fair
**Rating:** 5
**Confidence:** 5

**Summary:**

The authors propose a multi-level contrastive learning (MCL) method tailored for dense prediction downstream tasks. The core idea is to downsample original images in different scales and assemble them in a montage style as input. Since the location of each sub-image is known as a priori, the authors use this supervision signal to perform contrastive learning on multi-scale features. Extensive experiments demonstrate that the proposed method achieves superior performance when transferred to various dense prediction tasks.

**Strengths:**

-	The idea of composing images in a montage style is interesting.
-	The paper is generally well-written and easy to follow.
-	The downstream evaluation experiments are extensive and the results seem promising.

**Weaknesses:**

-	**Effectiveness on non-object-centric datasets.** In the introduction section, the authors claim that object-level SSL methods use off-the-shelf algorithms to produce object proposals, which are not accurate enough on non-object-centric datasets like COCO. To tackle this issue, the authors should mainly conduct pre-training experiments on COCO to compare with previous object-level SSL methods. However, in the paper, the authors mainly pre-train MCL on the object-centric ImageNet dataset. From my understanding, MCL still relies on the object-centric bias of ImageNet since the proposed montage strategy should ensure that each sub-image only contains a single object. Otherwise, pulling sub-images containing multiple objects together will mislead the network. Therefore, I am afraid that the proposed montage strategy may not be suitable for non-object-centric datasets like COCO. According to Table 1 (the only pre-training experiments on COCO in the whole paper), the COCO-pretrained MCL (41.8 $AP^{bb}$, 37.7 $AP^{mk}$) performs worse than the ImageNet-pretrained MCL (42.5 $AP^{bb}$, 38.2 $AP^{mk}$). Considering that the authors have adjusted the number of epochs to match the training iterations on ImageNet, such results further indicate that MCL may not be effective on non-object-centric datasets.

-	**Potentially unfair comparisons.** Apart from the backbone, the authors also pre-train FPN neck and detection head. However, most previous methods do not pre-train FPN and detection head. To ensure a fairer comparison, the authors should also pre-train FPN and detection head for previous methods (e.g., DenseCL). Another option is to pre-train MCL without FPN and detection head. This can better verify whether the montage assembly itself is superior over other pixel-level or object-level pretext tasks without the interference of other factors. According to Table 10 (e), only pre-training the backbone (41.1 $AP^{bb}$, 37.2 $AP^{mk}$) does not perform better than many previous methods that also only pre-train the backbone, as shown in Table 3. Given the current results, I am concerned that the main gains come from using additional multi-scale features in FPN for contrastive learning rather than the montage assembly. Furthermore, the claim that MCL can reduce pre-training epochs may be simply due to pre-training FPN and detection head. In other words, previous methods may also be able to reduce pre-training epochs by pre-training FPN and detection head.

-	**Some related works are missing.** There are also several closely related dense/object-level SSL works (e.g., [1, 2, 3]) that should be discussed and compared in Sec. 2 and Sec. 4.

References:

[1] Self-Supervised Visual Representation Learning by Contrastive Mask Prediction. In ICCV, 2021.

[2] Unsupervised Object-Level Representation Learning from Scene Images. In NeurIPS, 2021.

[3] Point-Level Region Contrast for Object Detection Pre-Training. In CVPR, 2022.

**Questions:**

I have additional questions:

-	The authors use a large batch size of 4096 during pre-training. Considering that the number of composed images is correlated with the batch size, I am curious about the effect of the batch size on MCL. Whether a small batch size can also work well?

-	The authors use a contrastive loss with positive and negative samples. Considering that many recent works (e.g., BYOL, SimSiam) remove the necessity of negative samples, what if a similar loss without negative samples is used in MCL? Will the performance be degraded?

-  The authors use an additional predictor for the online network. Could the authors provide a justification on this?

-	I am confused about the setting that extends MCL to the supervised learning scenario. According to the implementation details in the main text and the supplementary material, if I understand correctly, all models use supervised pre-training, while MCL is only used together with supervised fine-tuning. If this is the case, other dense/object-level pretext tasks (the authors could choose some baselines in Table 3 for comparisons, e.g., SoCo) should also be implemented in this setting to demonstrate the superiority of MCL.

Some other suggestions:

-	Sec. 4.2: Pre-training on Non-object-centric Dataset: "1.4 $AP^{bb}$/1.3 $AP^{mk}$ over SoCo" -> "1.2 $AP^{bb}$/1.3 $AP^{mk}$ over SoCo".

-	Some numbers in Table 1 are not correct. For example, Self-EMD $AP^{bb}$ should be +1.5 rather than +1.3, SoCo $AP^{bb}$ should be +1.7 rather than +1.5. Please also check other parts.

---

### Official Review · Reviewer_rYNZ · 2023-10-30

**Soundness:** 3 good
**Presentation:** 3 good
**Contribution:** 2 fair
**Rating:** 6
**Confidence:** 4

**Summary:**

The paper proposes a multi-level contrastive method for dense prediction task pretraining. The proposed multi-level contrastive method considers three key factors in dense prediction tasks: localization, scale consistency, and recognition. The author performs extensive experiments on the COCO dataset. The experiment results show its superiority compared with previous methods.

**Strengths:**

i) The motivation is reasonable and the writing is good

ii) The proposed method conducts extensive experiments to verify its effectiveness. The experiment results seem good and outperform several related works.

iii) The ablation study is sufficient, which discusses the impact of different sub-modules.

**Weaknesses:**

i) The idea of introducing multi-level features in contrastive learning to improve dense prediction tasks is not novel. It has been used in several previous works.

ii) The author thinks that previous contrastive learning only considers image-level feature alignment. However, in MCL, the feature of sub-image feature in the montage assembled image is also an image-level feature. I think MCL does not really apply object-level feature alignment.

**Questions:**

Refer to the questions.

---

### Official Review · Reviewer_yNPS · 2023-10-31

**Soundness:** 3 good
**Presentation:** 3 good
**Contribution:** 2 fair
**Rating:** 5
**Confidence:** 5

**Summary:**

This paper propose to stitch images with / without objects of different scales into a large image and thus conduct contrastive pre-training at different feature scales to improve the efficiency of CL pretraining (MCL), which enables the neural network to learn regional semantic representations for translation and scale consistency. They also adopt a scale-aware positive target assignment strategy on the feature pyramid to increase semantic feature representations. MCL consistently outperforms previous contrastive methods on datasets like COCO.

**Strengths:**

* Previous works have indicated that multi-positive contrastive training can greatly improve the training efficiency of SSL, this paper find a way to incorporate mutli-scale information into this procedure by stitching multiple images and arange their scales properly to achive the goal.
* The paper is clearly written.
* According to "Rethinking ImageNet Pre-training" of Kaiming He, Faster R-CNN with rand init and 6x schedules can achieves 41.3 box AP, and mask r-cnn generally brings about ~1 AP to detection, which shoud be around 42.3 AP. The proposed pre-training method is effective considering that it surpasses the upper bound of supervised training with enough epochs.

**Weaknesses:**

* The idea of stitching images together to improve the detection has been verified in paper "Dynamic Scale Training for Object Detection" of Yukang Chen, which is basically the same as this paper, so I doubt the novelty in this paper MCL, although it is disigned for pretraining of object detection. I think then shares similar motivations.
* Most of previous contrastive SSL works uses Faster R-CNN as downstream baseline, not Mask R-CNN. So I think the author should also report results on Faster R-CNN for fair comparison.
* Most of the references are at year 2020 and 2021, more recent works should be included considering that 2024 is only few day way.

**Questions:**

* Can the proposed strategies can also work on transformer architectures? I thinks results using MoCov3 / DINO should also be reported.
* Could the stitching strategy can be applied to more advanced pre-training startegies like MAE?

---

### Official Review · Reviewer_EixD · 2023-11-08

**Soundness:** 2 fair
**Presentation:** 3 good
**Contribution:** 2 fair
**Rating:** 3
**Confidence:** 5

**Summary:**

This paper introduces a self-supervised learning method named Multi-Level Contrastive Learning (MCL) for dense prediction tasks. MCL constructs pyramid-structured input images by stitching images scaled to different resolutions and conducts contrastive learning on the output feature pyramid to complete pre-training. MCL achieves impressive performance on downstream dense prediction tasks.

**Strengths:**

- Expanding the dataset by performing scaling and stitching operations at the raw data level is an interesting concept, and the interaction between different sub-images within the same input image is intriguing.

- This paper has conducted extensive experiments across a wide range of downstream tasks.

- This paper is well-written and logically structured.

**Weaknesses:**

- The idea of Montage Assembly shares similarities with MultiCrop[1], which constructs inputs of two different resolutions and performs contrastive learning on the output features. Montage Assembly, however, constructs a montage of multiple resolutions. Thus, an initial analysis should consider the effect of stacking images in the batch dimension instead of the spatial dimension, retaining structure to obtain the same features for contrastive learning, and whether this leads to a difference in effectiveness. If there is a difference, given that stacking images spatially is also an unusual operation, what could be the reason for this difference?

- This paper seems to underreport the results of other works.
    - In their original paper, DetConB can reach a bbox AP of 42.7, and SoCo can reach 43.2, but it seems that only the lower results from shorter training epochs are reported in Table 2.
    - The semi-supervised results in Table 4 seem to be significantly lower than those reported in SoCo, yet the results of SoCo are not reported in this table.
    - In Table 5, the results of SoCo are significantly lower than those reported in their paper.

- Considering that montage assembly introduces additional input images in each iteration, MCL could indeed have a greater computational overhead compared to methods that do not increase the number of input images. If this is the case, it would be necessary to quantify the specific additional computational cost. This information is crucial for evaluating the efficiency of the MCL method against other approaches.

[1] Unsupervised learning of visual features by contrasting cluster assignments

**Questions:**

Please refer to the weaknesses.